Identification and biological characterization of pathogen causing sooty blotch of Ardisia crispa (Thunb.) A.DC.

Yang Demei 1
Luo Jiangli 1
Zhou Ying 1 yingzhou71@sina.com
Zhou Sixuan 2
Liu Xiongwei 1
Liu Chang 1 19liuchang@163.com
1 School of Pharmacy, Guizhou University of Traditional Chinese Medicine , Guiyang , China
2 Institute of Animal Husbandry and Veterinary Sciences of Guizhou Province , Guiyang , China
Mata Fernando
Electronic publication date: 2025 Mar 24
Publication date: 2025
Volume: 13
Electronic Location ID: e19130
Received 2024 Nov 8; Accepted 2025 Feb 18
Copyright: © 2025 Yang et al.
Copyright year: 2025
Copyright holder: Yang et al.
License: This is an open access article distributed under the terms of the Creative Commons Attribution License, which permits unrestricted use, distribution, reproduction and adaptation in any medium and for any purpose provided that it is properly attributed. For attribution, the original author(s), title, publication source (PeerJ) and either DOI or URL of the article must be cited.
License URL: https://creativecommons.org/licenses/by/4.0/

Keywords: Ardisia crispa (Thunb.) A.DC., Sooty blotch, Annulohypoxylon stygium, Diaporthe angelicae, Biological characteristics, Biocontrol mechanism

Funding: Science and Technology Fund of Guizhou Province QKHJC-ZK [2024] general 376 Graduate education Innovation program of Guizhou University of Traditional Chinese Medicine YCXKYS2023010 High-level Innovative Talents of Guizhou Province (QianKeHe platform talents-GCC [2023] 047) This work was funded by the Science and Technology Fund of Guizhou Province (QKHJC-ZK [2024] general 376); Graduate education Innovation program of Guizhou University of Traditional Chinese Medicine (YCXKYS2023010); High-level Innovative Talents of Guizhou Province (QianKeHe platform talents-GCC [2023] 047). The funders had no role in study design, data collection and analysis, decision to publish, or preparation of the manuscript.

==============================
Sooty blotch, a primary leaf disease affecting Ardisia crispa (Thunb.) A.DC. (A. crispa, AC), significantly impacts both the yield and quality of this medicinal plant. However, the specific species of pathogenic fungi responsible for this disease and their mechanisms of pathogenesis remain unclear. To elucidate the etiology of sooty blotch, it is essential to investigate effective prevention and treatment methods, and provide a theoretical basis for the effective protection of AC. Initially, the alterations in internal organelles that result in sooty blotch were examined using transmission electron microscopy (TEM) and scanning electron microscopy (SEM). Additionally, differential genes were analyzed using differential display reverse transcriptase-PCR (DDRT-PCR) in both healthy and diseased leaves of AC. Moreover, the pathogenic fungi were separated, purified and identified from leaves infected with sooty blotch of AC, and subsequently, their pathogenicity and biological characteristics were tested. Furthermore, the inhibitory effect of pathogens was measured using the water extract of traditional Chinese medicine, based on the growth rate of the mycelium. The findings indicated that the photosynthesis rate of diseased leaves was slower than that of healthy leaves, as revealed by TEM and SEM analyses. Additionally, DDRT-PCR results demonstrated that the differentially expressed genes primarily included those related to zinc finger proteins, acyl-CoA-transferases, and chloroplasts. The phylogenetic tree and pathogenicity test results showed that the pathogens causing sooty blotch of AC were Annulohypoxylon stygium and Diaporthe angelicae. Microscopic observation revealed that Annulohypoxylon stygium and Diaporthe angelicae exhibited distinct microscopic characteristics, and a pH range of 7–10 along with a subdued light environment were more conducive to the growth of pathogens. Additionally, the water extract of Lonicera fulvotomentosa Hsu et S. C. Cheng and A. crispa (Thunb.) A.DC. had a strong antifungal action on the two pathogens of sooty blotch, and the water extract of Ardisia crenate Sims had a better antifungal action on the Diaporthe angelicae. In this study, Annulohypoxylon stygium and Diaporthe angelicae were reported as pathogenic fungi causing sooty blotch for the first time, and affected the photosynthesis of AC leaf, and these study provides a theoretical basis for the diagnosis and prevention of A. crispa (Thunb.) A.DC. sooty blotch.

Introduction

Sooty blotch is a widespread fungal disease affecting a variety of plant species, including Chinese medicinal materials, fruit trees, crops, flowers, and so on. A black mold develops on the infected areas of the plant, which impedes photosynthesis and leads to weak growth, early leaf drop, and primarily affects the leaves, and then the disease can also spread to the branches and eventually cause the plant to wilt gradually (Tham et al., 2023). The pathogens of sooty blotch are typically spread through rain or air currents, carried as spores or fragments of mycelium. When these spores or mycelia germinate on substrates containing honeydew, the sooty blotch pathogenic fungi colonize the plant surface. Initially, the affected plant surface may produce black, bituminous material or have a fluffy center. Over time, the mildew spreads and forms a layer that covers the entire leaves, shoots, and fruits. In the advanced stage of the disease, pathogenic fungi produce structures such as ostracods and conidia in the bituminous coal. These structures are the primary source of infection in the second year after overwintering, leading to multiple re-infections (Gleason et al., 2019). This leads to the loss of crops and medicinal plant, the decline of economic output, the degradation of landscape functions of garden plants, and even environmental pollution. The disease is highly contagious and becomes even more so as it spreads. It has inflicted severe damage on orchards, medicinal bases, and nursery lands (Araujo et al., 2021).

Radix Ardisia, a valued Miao herbal remedy in Guizhou, is known for its laryngeal benefits (Liu et al., 2022). In 2021, Kaihou Jian Spray, containing this herb, reached 880 million yuan in sales, reflecting rising market demand for its base plants. Ardisia crispa (Thunb.) A.DC., the botanical source of Radix Ardisia, is a medicinal plant with a history in traditional Chinese medicine (Wu et al., 2025). It contains alkaloids, saponins, and polysaccharides, which provide unique pharmacological effects such as clearing throat, removing phlegm and dampness, dispersing stasis and eliminating swelling (Wu et al., 2025). It’s used for treating lung heat, cough, phlegm production, and asthma. A. crispa (Thunb.) A.DC. also has anti-inflammatory and immunomodulatory properties, attracting modern medical research. However, A. crispa (Thunb.) A.DC. is prone to sooty blotch in Guizhou’s humid climate, which can lead to black mildew on leaves, hindering photosynthesis and potentially causing plant death. The increasing incidence of this disease negatively impacts yield and quality, causing economic losses.

Sooty blotch, a global issue, hosts various pathogenic fungi. Research shows 37 genera of pathogens are linked to it, with Zygophiala, Microcyclospora, Stomiopeltis, and Ramichloridium being some of these (Zhang et al., 2019). These fungi are adaptable and stress-resistant, surviving in different climates and soils. To manage sooty blotch diseases, it’s crucial to study these fungi. Understanding their growth, reproduction, transmission, and pathogenicity is key for targeted prevention and control strategies. Currently, chemical pesticides like carbendazim and chlorothalonil are used, but they can harm the environment, lead to resistance, and reduce effectiveness. Biological control, on the other hand, offers pollution-free benefits and sustained control without resistance. It’s a vital component for sustainable development. However, there’s limited research on biological control of sooty blotch in A. crispa (Thunb.) A.DC. Comprehensive research is needed to provide new insights and strategies for managing this disease.

In this study, the changes of organelles were utilized to clearly observe in the leaves of A. crispa (Thunb.) A.DC. under normal and infected conditions through transmission electron microscopy (TEM) and scanning electron microscopy (SEM). To analyze the differentially expressed genes in the leaves of A. crispa (Thunb.) A.DC. using DDRT-PCR at the molecular level, the aim is to understand the response mechanism to sooty blotch. Additionally, typical pathogenic fungi responsible for sooty blotch were isolated and purified to elucidate the types of pathogenic fungi and the pathogenesis of sooty blotch. The microscopic characteristics of the pathogenic fungi were identified using microscopy. The pathogenicity of the isolated fungi was observed, and their morphological and molecular biological characteristics were investigated. Concurrently, the inhibitory effects of eight types of traditional Chinese medicine on pathogenic fungi were assessed using the mycelium growth rate method, leading to the selection of appropriate antifungal agents. The study elucidates the molecular mechanisms of biological action from a genetic perspective, offering a reference for related research on the prevention and control of sooty blotch. It contributes to enhancing the quality and output of Chinese medicinal materials, enriches the disease research of A. crispa (Thunb.) A.DC. sooty blotch, and facilitates more in-depth subsequent research.

Materials and Methods

Materials

Normal and disease leaves of Ardisia crispa (Thunb.) A.DC. (A. crispa, AC) were collected from the Guizhou University of Traditional Chinese Medicine (N.109.437569, E.19.19680); Professor Jie-hong Zhao from the Guizhou University of Chinese Medicine, identified the infected leaves as sooty blotch leaves. High speed refrigerated centrifuge Sorvall Legend Micro 17R (Thermo Fisher Scientific). Polysaccharide polyphenol plant total RNA extraction kit (Tiangen Biochemical Technology (Beijing) Co., LTD.), agar gel DNA recovery kit (Tiangen Biochemical Technology (Beijing) Co., LTD.), FastKing one-step genome cDNA first strand synthesis premix kit (Tiangen Biochemical Technology (Beijing) Co., LTD.). Fungal genome DNA extraction kit (Beijing Solaibao Technology Co., LTD.), 2xTaq PCR MasterMixII (Tengen Company).

TEM and SEM analysis

Transmission electron microscopy (TEM) was performed using the following optimized protocol: the sample was prefixed with 3% glutaraldehyde and then 1% osmium tetroxide. To ensure dehydration, the concentration gradient of the dehydrating agent acetone is 30%→50%→70%→80%→90%→95%→100%. Then, the samples were permeated and embedded using a mixture of dehydrating agent and Epon812 embedding agent. The microtome was used to generate ultrathin sections with a thickness of 60 to 90 nm. The uranium acetate and lead citrate was performed to staining, and the images were gathered and analyzed to observe specific lesions. Scanning electron microscope (SEM): ultra-pure water was washed twice, 5 min each time, and the concentration gradient was 30%→50%→70%→80%→90%→95%→100% after concentration dehydration of ethanol series, and the specific lesions were observed after ion sputtering 10 min each time.

DDRT-PCR detection

The selection process involves several critical factors, including primer length, GC content, annealing temperature, and the avoidance of primer dimer formation. Typically, primer lengths range from 18 to 24 base pairs to ensure sufficient specificity and binding strength.The GC content should be maintained between 40% and 60% to ensure the stability of the primer and its annealing efficiency. The annealing temperature is a crucial determinant for the effective binding of primers to templates; thus, preliminary experiments are essential to evaluate both the amplification efficiency and specificity of these primers, in order to select those most suitable for experimental requirements. The total RNA was derived from AC-normal and disease leaves according to RNA extraction kit. The first cDNA was synthesized by using a Fastking one-step method, excluding the leading strand synthesis premixed reagent. Using the normal and disease cDNA as template, the PCR reaction system consisted of 1 µL cDNA, 1 µL random primers, 1 µL anchored primers, 10 µL 2xPCR mix, 7 µL ddH2O. The amplification procedures were as follows: 94 °C 5 min, 94 °C 40 s→42 °C 40 s→72 °C 90 s (40 cycles), 72 °C extension for 10 min. See anchored primers and random primers sequences in Tables S1 and S2.

The PCR samples were loaded onto a 6% polyacrylamide gel, subjected to electrophoresis, and then photographed and recorded using a gel imaging system. The differential bands were excised and placed in a PCR tube containing 100 µL of enzyme-free sterile water. The mixture was subsequently crushed into a fine powder and incubated in water at 85 °C for 15 min. Following centrifugation at 12,000 r/min for 15 min, 3 mol/L NaAc (pH 5.2) and 2.5 times the volume of anhydrous ethanol were added to 1/10 the volume of the supernatant to precipitate the DNA. The mixture was centrifuged again for 15 min at 12,000 r/min. The precipitate was washed once with 70% ethanol and once with anhydrous ethanol, each time centrifuging for 15 min. Finally, the DNA was dried and dissolved in 20 µL DEPC water. 1 µL template was used only for re-amplification, and the amplification system used was similar to that described above. The re-amplified DNA fragments were identified by 1% agarose gel electrophoresis. Marker D2000 was invoked as the control, and the electrophoresis conditions were set to 120V voltage, 400 mA current, and 30 min duration. The DNA was then recovered using an agarose gel DNA recovery kit and subsequently detected by 1% agarose gel electrophoresis. The recovered DNA was further amplified by PCR using the same a re-amplified DNA fragments were visualised in a 1% agarose gel mplification system as mentioned earlier. The 40 µL amplification product was then collected for sequencing. The obtained sequences were tested and compared for similarity with different fragments and related genes using BLAST tools. Additionally, the function of gene searching was associated with Ensembl, Gene-Card, and other databases for comprehensive analysis.

Isolation and purification of pathogenic fungi

The infected leaves were washed with clean water and cut into 5 mm × 5 mm sizes, and make sure to remove external contaminants. The retained parts were cleaned with sterile water and then shaken with 75% ethanol for 30 s. Afterwards, they shaken with 1% NaClO for 6 min. The parts were washed with sterile water for 2–3 times and then cultured in Sand’s solid medium for 5–7 days, followed by purification for 2–3 times.

Molecular identification

Molecular identification encompasses DNA sequence analysis, genome comparison, and the application of molecular markers. Through the integration of these genetic traits, it becomes possible to accurately distinguish and identify various fungal strains. The Fungal Genomic DNA (Soleibao) extraction kit was used to extract DNA, and 1% agarose gel electrophoresis was employed to detect the DNA with photos taken in the gel imaging system. For PCR amplification, 2 µL DNA was used as the template and mixed with 8.2 µL water, 10 µL Mix enzyme, and 0.4 µL upstream and downstream primers (Table S3). The PCR amplification was performed with the following conditions: initial denaturation at 94 °C for 5 min, followed by 35 cycles of denaturation at 94 °C for 30 s, annealing at 55 °C for 30 s, and extension at 72 °C for 30 s. Finally, extension was performed at 72 °C for 10 min. The PCR products were subsequently sequencing, and the pathogenic fungi gene sequences were analyzed using a Basic Local Alignment Search Tool (BLAST) platform in the NCBI database. Based on the identification and screening of reference strains, and MEGA11.0 software was used to laborately constructed phylogenetic tree.

Pathogenicity test of pathogenic fungi

The leaf mycelium inoculation method was employed to treat the normal leaves of AC in vitro, and normal leaves were used in the blank control experiment. The leaves were carefully placed on a sterilized filter paper in a flat plate under aseptic conditions on a clean table. Then, the mycelium was applied onto the leaf surface, and sterile water was added, ensuring that the water level did not exceed half of the leaves. It was cultured at a consistent temperature and humidity of 28 °C for a duration of 12 days. Throughout the culture period, the development of leaf disease was observed and the symptoms and characteristics were carefully recorded and photographed. The mycelium morphology of pathogens was observed under a microscope. It includes characteristics of fungal colonies, the shape and size of spores, and the structure of mycelia.

Biological characteristics of pathogenic fungi

Effect of pH on pathogenic fungi growth

The effects of pH on the growth of pathogenic fungi were investigated using Scharr’s culture medium. Eight different pH levels (5, 6, 7, 8, 9, 10, 11 and 12) were ready. To create holes for inoculation, a 5 mm hole punch was utilized to drill holes in the edge of the pathogen colony. The pathogenic fungi cake was then inoculated into the holes, and each group was repeated three times. The cultures were incubated in an incubator at 28 °C. After 6 days of incubation, the colony diameter was monitored using the cross-crossing method.

Effect of light on pathogenic fungi growth

The effects of light on the growth of pathogenic fungi were examined in this study. The 5 mm hole punch was utilized to create holes at the periphery of the pathogenic fungi colony. The pathogenic fungi were cultured on Scharr’s culture medium, and three different light conditions were tested: 24 h of continuous light, 12 h of light pursued by 12 h of darkness, and 24 h of complete darkness. Each treatment group was replicated three times, and the cultures were kept at a constant temperature of 25 °C. After 6 days of incubation, the colony diameter was detected by the cross-crossing method.

Biological control of pathogenic fungi

With water as solvent, eight kinds of traditional Chinese medicine, containing Ardisia crispa (Thunb.) A.DC., Lonicera fulvotomentosa Hsu et S. C. Cheng, Alangium chinense (Lour.) Harms, Lonice Raejaponicae Caulis, Sophorae Tonkinensis Radix et Rhizoma, Ardisia crenata Sims, Aletris spicata (Thunb.) Franch, Persicaria capitata (Buch.-Ham. ex D.Don) H.Gross were extracted by reflux heating extraction method, combined with filtrate, heated and concentrated to 50 mL (1 g/mL), and stored at 4 °C for later use. A 25 mL concentrated Chinese medicine solution was mixed with 225 mL medium (100 mg/mL) and sterilized at 121 °C for 20 min to prepare the medicated medium. Good active cake with a diameter of 4 mm was drilled with a hole punch and placed in the center of the medicated medium. The positive control was 200 µg/mL for carbendazim and the blank control was sterile water. Each Chinese medicine water extract and pathogenic bacteria, positive control and blank control were repeated in three groups. After incubating at a constant temperature of 28 °C for 6 days, the colony diameter was monitored using the cross method. The experiment was conducted three times for each group, and the colony diameter of each group was measured using the cross-hatching method to obtain three sets of data. Subsequently, the average colony diameter and antibacterial rate for each group were calculated. The mean values and antibacterial rates were determined, along with the standard deviation for each dataset. Calculate the inhibition rate [(colony diameter of the blank group - colony diameter of the treatment group)/(colony diameter of the blank group - 0.4) * 100%].

All data are presented as mean ± SD (standard deviation of the mean) and analyzed using SPSS 27.0 software. For three or more groups, one-way analysis of variance (ANOVA, p < 0.05) was applied (Chatzi & Doody, 2023; Jan & Shieh, 2014). Prior to conducting the ANOVA, the data was verified to meet the necessary assumptions. Specifically, the normality of the data was assessed using the Shapiro-Wilk test and by visually inspecting Q-Q plots. Additionally, the test for homogeneity of variances was conducted using Levene’s test. For data with homogeneity of variance and normal distributions, with adjustments for multiple comparisons using Tukey’s Honest Significant Difference (HSD) test for post-hoc comparisons, as it is appropriate for controlling Type I error rates in multiple comparisons. For data with unequal variances and non-normal distributions, post-hoc comparisons should follow the Kruskal-Wallis one-way ANOVA, utilizing the the Tamhane and Dunn’s test with Bonferroni correction to control Type I errors. The significant letter (a, b, c, d, e .. .. ..) marking method was used to indicate significant differences among multiple sets of data. Different lowercase letters for the same indicator represent significant differences (p < 0.05), and the same lowercase letters represent non-significant differences (p > 0.05).

Results

The results of transmission electron microscope and scanning electron microscope

Under transmission electron microscopy (TEM), the chloroplasts in the normal leaf mesophyll cells of Ardisia crispa (Thunb.) A.DC. (A. crispa, AC) were neatly arranged with clear and complete structures, and they contained a few starch granules. The cell membrane was discontinuous, and the cytoplasmic contents were dispersed in most regions (Fig. 1A). In the mesophyll cells of the AC-infected leaves, there were few chloroplasts, and most appeared wrinkled, circular in shape, and small in volume. The cytoplasm contained a small number of vacuoles and lipid droplets. Under electron microscopy, large vacuoles were suspected to contain free cytoplasm, and a large number of bacteria were also observed in the triangular area, with discontinuous cell membranes in some regions (Fig. 1B).

Figure 1 The results of transmission electron microscope observation.

(A) Healthy leaves; (B) disease leaves.

Under scanning electron microscopy (SEM), the apparent morphology of the front, reverse and section of the normal and infected leaf was uncommon. There were close-arranged and complete upper epidermal cells on the front of AC-healthy leaves, and the pores in the lower epidermis are uniform in size and orderly in arrangement, and most of them are concealed pores on the reverse side of the healthy leaves (Fig. 2A). The AC-healthy leaves section showed that the internal structure was transparent, the palisade tissue was arranged neatly, and the spongy tissue and vascular sheath cell tissue were normal. The helical mycelia were noted to be located between the mesophyll cells of the AC-disease leave (Fig. 2B). These mycelia produced haustoria on the mesophyll cells, which greatly enhanced their ability to connect to the cells. Additionally, the mycelium was observed to protrude from the stomata on the opposite sides of the leaves. In the leaf cross-section, spiral advancing mycelia were observed to traverse through the cell space, while also protruding from the stomata on the back side of the leaf.

Figure 2 The results of scanning electron microscope observation.

(A) Healthy leaves; (B) disease leaves.

DDRT-PCR detection results and analysis

Total RNA was removed from both AC-healthy and AC-disease leaves and subjected to 1% agarose gel electrophoresis for detection. The bands observed were distinct, with the 28 S band appearing approximately twice as bright as the 18 S band, which detects a high level of RNA integrity (Fig. S1). Three anchoring primers B0327, B0328 and B0329 were combined with 26 random primers B0301~B0326 for PCR amplification, and 199 differential bands were isolated by polyacrylamide gel electrophoresis (Fig. S2). A total of 150 different bands were obtained from AC-disease leaves samples, and 49 discrete bands were obtained from AC-healthy leaves samples, with the size ranging from 100 to 2,000 bp, which was consistent with the rule of difference display band results. After DNA recovery from 199 different bands by agarose gel electrophoresis (Fig. S3), a total of 48 single bands with strong specificity were obtained (Fig. S4). The sequencing results were compared using BLAST in the NCBI database, leading to a total of 69 sequences related to the study (Table S4). The gene sequences were annotated using gene-bank, and the detailed results can be noted in Table S5. The subsequent homology analysis revealed the involvement of six different metabolic pathways in the samples, namely: (1) stress resistance; (2) photosynthesis; (3) cell respiration; (4) cell proliferation; (5) antioxidant; and (6) Intracellular transport.

Molecular identification results and analysis

A total of 11 pathogenic fungi strains were obtained from infected leaves of AC. Following phenotypic identification, the strains were appointed as AC1, AC2, AC3, AC4, AC5, AC6, AC7, AC8, AC9, AC10 and AC11 (Table S6), respectively. The strains were distinguished according to their morphological characteristics and identified by sequencing (Fig. 3; Table S7).

Figure 3 The colony of pathogenic fungi.

AC: Ardisia crispa (Thunb.) A.DC.

Pathogenicity test results and analysis

The disease morphology of Annulohypoxylon stygium (AC10) is different in different periods, the mycelium grew up obviously on the third day, a small part of the leaves began to turn yellow on the eighth day, the leaves are yellowish in whole and black spots can be seen all over the mycelium on the 12th day. The disease morphology of Diaporthe angelicae (AC11) in different periods is various, the mycelium grew up obviously on the third day, the tip of the leaf turned yellow on the eighth day, the mycelium spread long foliar surface and most of the foliar surface turned yellow on the 12th day (Fig. S5).

The morphological characteristics and the sequencing results showed that the pathogenic fungi of sooty blotch were Annulohypoxylon stygium and Diaporthe angelicae (Fig. 4). Microscopic observations revealed distinct microscopic characteristics among diverse pathogenic fungi of sooty blotch. The mycelium of Annulohypoxylon stygium exhibited a thick wall, large curvature, few branches, and a slender clumpy and separated appearance. On the other hand, the hypha of Diaporthe angelicae displayed a thin transparent wall, low curvature, abundant branches, and slender clusters (Fig. 4).

Figure 4 Observation of microscopic characteristics of pathogenic fungi of sooty blotch (10 × 40).

The microscopic morphology of the two pathogenic fungi was observed to distinguish the difference from that of other endophytic fungi.

The phylogenetic tree was analyzed the evolutionary relationships of Annulohypoxylon stygium (Fig. 5A), Diaporthe angelicae (Fig. 5B) and other relevant strains. The topology of phylogenetic trees presents an interesting insight into genetic differences. Notably, AC10 and AC11 were respectively closely related to MG881822.1 and KC343029.1, and indicating a common descent. These observation strongly suggests that there are significant genetic differences between AC10, AC11 and their congeners. The results suggest that phylogenetic trees offer powerful understanding about the evolutionary relationships of pathogenic fungi and their underlying genetic relationships with reference strain.

Figure 5 Phylogenetic tree.

(A) Homology comparison between AC10 and related species; (B) homology comparison between AC11 and related species.

Results and analysis of biological characteristics of pathogenic fungi

Results and analysis of the effect of pH on the growth of pathogenic fungi

pH testing results revealed that both pathogenic fungi were able to grow within a pH range of 5 to 12 (Table S8). However, the mycelial growth rate of AC10 was significantly slower at pH 6 and below, indicating that an overly acidic environment was not suitable for its growth. Conversely, the mycelial growth rate of AC10 was faster within a pH range of 7 to 12, with no significant difference found between them. This indicates that the pathogenic fungi is more suitable for growth in a neutral or slightly alkaline environment. The growth pattern of AC11 mirrored that of AC10 (Table S9). The colony morphology of both pathogenic fungi on the sixth day of growth is shown in Fig. 6A, while the colony diameters at different pH values on the sixth day in Figs. 6B, 6C. If the entire plate is overgrown, the colony diameter is recorded as 8.60 cm.

Figure 6 Positive and negative colony morphology and colony diameter of the two pathogenic fungi on the 6th day of growth.

(A) Growth state of AC10 and AC11 under different pH conditions; (B) average colony diameter of AC10 at different pH; (C) Average colony diameter of AC11 at different pH.

Results and analysis of the influence of light on the growth of pathogenic fungi

The photoperiod test results revealed distinct effects of different light conditions on the pathogenic fungi. AC10 showed greater growth in full-bright conditions (Table S10), while there was no significant difference in mycelial growth rate between full darkness and half-light/half-darkness conditions. However, the difference in growth rate was more pronounced compared to maximum light conditions. AC11, on the other hand, was not influenced by light (Table S11). The colony morphology of the two pathogenic fungi on the 6th day of growth and the colony diameter under different light conditions on the 6th day are shown in Fig. 7. In the case of overgrowth of the entire plate, a diameter measurement of 8.60 cm was recorded.

Figure 7 Colony morphology and colony diameter of the two pathogenic fungi on the 6th day of growth.

(A) Growth state of AC10 and AC11 under different light conditions; (B) average colony diameter of AC10 under different light conditions; (C) average colony diameter of AC11 under different light conditions.

Biological control results and analysis

It can be seen that different Chinese herbal extracts had different inhibitory effects on pathogenic fungi at the same concentration. The results showed that the inhibitory rate of different plant extracts against Annulohypoxylon stygium was significantly different (Table S12), the normality of the data was assessed using the Shapiro-Wilk test and visual inspection of Q-Q plots. The data that met the assumptions of normality and homogeneity of variances, one-way ANOVA followed by Tukey’s HSD post-hoc test was used to compare groups, different lowercase letters for the same indicator represent significant differences (p < 0.05), and the same lowercase letters represent non-significant differences (p > 0.05) (Table 1). For the pathogenic fungi Annulohypoxylon stygium, antifungal rate was Ardisia crispa (Thunb.) A. DC. ( 62.1633 ± 2.8403) > Lonicera fulvotomentosa Hsu et S. C. Cheng (54.3133 ± 0.6982) > Alangium chinense (Lour.) Harms (36.8733 ± 3.2520) > Lonice Raejaponicae Caulis (33.7200 ± 0.2300) > Sophorae Tonkinensis Radix et Rhizoma (32.6433 ± 4.0539) > Ardisia crenata Sims ( 28.2500 ± 1.8644) > Aletris spicata (Thunb.) Franch ( 6.6233 ± 2.0842) > Persicaria capitata (Buch.-Ham. ex D.Don) H.Gross (−13.3167 ± 2.7936). In summary, the water extract of A. crispa (Thunb.) A.DC. and Lonicera fulvoto-mentosa Hsu et S. C. Cheng exhibits potent inhibitory effects against Annulohypoxylon stygium. The inhibitory rates for Ardisia crenata Sims, Lonice Raejaponicae Caulis, Sophorae Tonkinensis Radix et Rhizoma, and Alangium chinense (Lour.) Harms are approximately 30%. The inhibitory effects of Aletris spicata (Thunb.) Franch are not significant, and Persicaria capitata (Buch.-Ham. ex D.Don) H.Gross may even promote their growth.

Table 1 The inhibitory rates of Chinese medicines to Annulohypoxylon stygium.

Traditional Chinese medicine	Hyphal diameter (cm)	Antifungal rate (%)	
Lonicera fulvotomentosa Hsu et S. C. Cheng	2.3783 ± 0.0301e	54.3133 ± 0.6982b	
Persicaria capitata (Buch.-Ham. ex D.Don) H.Gross	5.3067 ± 0.1210a	−13.3167 ± 2.7936e	
Ardisia crispa (Thunb.) A.DC.	2.0383 ± 0.1229f	62.1633 ± 2.8403a	
Ardisia crenata Sims	3.5067 ± 0.0808c	28.2500 ± 1.8644d	
Lonice Raejaponicae Caulis	3.2700 ± 0.0100cd	33.7200 ± 0.2300c	
Sophorae Tonkinensis Radix et Rhizoma	3.3167 ± 0.1756cd	32.6433 ± 4.0539cd	
Alangium chinense (Lour.) Harms.	3.1333 ± 0.1407d	36.8733 ± 3.2520c	
Aletris spicata (Thunb.) Franch	4.4433 ± 0.0902b	6.6233 ± 2.0842e	
Note:

The data are presented as the means ± SD (n = 3). Different lowercase letters for the same indicator represent significant differences (p < 0.05), and the same lowercase letters represent non-significant differences (p > 0.05).

For the Pathogenic fungi Diaporthe angelicae, antifungal rate was A. crispa (Thunb.) A. DC. (94.5367 ± 3.3270) > Ardisia crenata Sims (74.0433 ± 2.3027) > Lonicera fulvotomentosa Hsu et S. C. Cheng (49.5633 ± 7.3847) > Lonice Raejaponicae Caulis (32.5967 ± 2.3460) > Sophorae Tonkinensis Radix et Rhizoma (9.9200 ± 1.5216) > Alangium chinense (Lour.) Harms. (4.4267 ± 14.5974) > Aletris spicata (Thunb.) Franch (−0.5467 ± 4.3481) > Persicaria capitata (Buch.-Ham. ex D.Don) H.Gross (−34.4300 ± 0.0000). It can be seen that it had a good inhibitory effect on Diaporthe angelicae (Table S13), Lonicera fulvoto- mentosa Hsu et S. C. Cheng, A. crispa (Thunb.) A.DC., Ardisia crenata Sims, while Lonice Raejaponicae Caulis had a inhibitory rate of about 30%, Sophorae Tonkinensis Radix et Rhizoma and Alangium chinense (Lour.) Harms. had no obvious inhibitory effect, and Aletris spicata (Thunb.) Franch and Persicaria capitata (Buch.-Ham. ex D.Don) H.Gross had a slight promoting effect on growth. These data that did not meet the normality assumption, the non-parametric Kruskal-Wallis test was applied, followed by Dunn’s post-hoc test with Bonferroni correction, and different lowercase letters for the same indicator denote significant differences (p < 0.05), whereas the same lowercase letters signify non-significant differences (p > 0.05) (Table 2). The antifungal colony morphology of eight kinds of Chinese medicine aqueous extracts on two kinds of pathogenic fungi is illustrated in Fig. 8. This study provides a reference for the development of A. crispa (Thunb.) A.DC., Lonicera fulvotomentosa Hsu et S. C. Cheng and Ardisia crenata Sims as green antibacterial drugs for sooty blotch, so as to reduce the dependence on antibacterial drugs and the occurrence of resistance of pathogenic bacteria, and make the cultivation of Chinese medicinal materials develop in a more healthy and sustainable direction.

Table 2 The inhibitory rates of Chinese medicine to Diaporthe angelicae.

Traditional Chinese medicine	Hyphal diameter (cm)	Antifungal rate (%)	
Lonicera fulvotomentosa Hsu et S. C. Cheng	3.4767 ± 0.4506cd	49.5633 ± 7.3847 bc	
Persicaria capitata (Buch.-Ham. ex D.Don) H.Gross	8.6000 ± 0.0000a	−34.4300 ± 0.0000e	
Ardisia crispa (Thunb.) A.DC.	0.7333 ± 0.2031e	94.5367 ± 3.3270a	
Ardisia crenata Sims	1.9833 ± 0.1405d	74.0433 ± 2.3027b	
Lonice Raejaponicae Caulis	4.5117 ± 0.1429c	32.5967 ± 2.3460c	
Sophorae Tonkinensis Radix et Rhizoma	5.8967 ± 0.0950b	9.9200 ± 1.5216d	
Alangium chinense (Lour.) Harms.	6.2300 ± 0.8904ab	4.4267 ± 14.5974de	
Aletris spicata (Thunb.) Franch	6.5333 ± 0.2650ab	−0.5467 ± 4.3481de	
Note:

The data are presented as the means ± SD (n = 3). Different lowercase letters for the same indicator represent significant differences (p < 0.05), and the same lowercase letters represent non-significant differences (p > 0.05).

Figure 8 Positive and negative images of antifungal colony morphology of eight kinds of Chinese medicine water extracts on two kinds of pathogenic fungi.

Growth state of AC10 and AC11 in different medicated medium.

Discussion

Sooty blotch is a plant disease that is common in tropical and subtropical areas. As a broad concept for sooty blotch, suggesting that it should be collectively referred to as sooty blotch based on the outward forms of diseases caused by pathogenic fungi, rather than just trying to classify it (Wang, Zhang & Sun, 2021). The host range of sooty blotch is extensive, including most tropical trees, as well as some economic fruit trees and flower plants. Sooty blotch are widely distributed and occur in warm and humid areas of the world, and most of them are diverse. For example, at least 60 species of coal pollution bacteria have been reported in the United States, at least 12 genera and 30 species in China, at least 25 species of coal pollution bacteria in Spain, 19 species in Germany, six genera and 12 species in Turkey, seven genera in Norway, six genera and eight species in Poland, four genera and six species in Montenegro, and three genera and five species in Serbia and Slovenia (Li, Zhang & Sun, 2016 ), Mauritius, Hawaii, South Africa, Reunion Island, Pakistan, India, South China (Ganesan et al., 2022), and Taiwan. When sooty blotch occurs, the branches and leaves are the focus with a layer of mildew that resembles soot, hence its name. Sooty blotch indirectly impacts the plant’s physiological functions by reducing the effective photosynthetic area of the leaves, thereby altering the internal structure or composition of plant tissue. In the case of fruit trees, sooty blotch can also infect the fruits, causing black discoloration and decreasing their quality (Li et al., 2022). The harm caused by sooty blotch cannot be ignored (Liu et al., 2020). This common pathogenic fungi produces black colonies and spots on the surface of host plants, often damaging the leaves, stems, and fruits (Chomnunti et al., 2014). As a consequence, plant leaves may display black sooty mold layer symptoms on their upper surface.

Plant pathomycetes live on the surface of parasitizing plants and causing plant diseases. Pathogenicity tests revealed the presence of Annulohypoxylon stygium and Diaporthe angelicae as the two types of pathogenic fungi. Microscopic observation showed distinct microscopic characteristics in these diverse pathogenic fungi. Through the study of their biological characteristics, it was observed that the mycelium growth rate of the Annulohypoxylon stygium, which causes sooty blotch, was considerably slower at pH 6 and below compared to other groups. This indicates that an overly acidic environment is not suitable for the growth of pathogenic fungi (Li, Chen & Tian, 2022). The difference in growth rate was considerably larger when compared to the growth rate under full-light conditions. Among the approximately 1.0 × 105 species of fungi currently-known, more than 8,000 species of fungi can cause diseases on plants (Sang & Kim, 2020). After plant tissues are adversely affected by pathogenic fungi, the defense enzyme system in the host body will undergo corresponding changes (Rajani et al., 2021). In the incompatibility interaction, local cell necrosis plaques significantly different from those surrounding healthy tissues are constituted at the infected site, that is, hyper-sensitive reaction (HR) (Li et al., 2022). In affinity interaction, some fungi use stomata or trauma on the host surface to invade and produce some infection structures formed by specialized mycelia. The pathogenic fungi can invade and establish a parasitic relationship with the host, and resulting in plant infection (Mapuranga et al., 2022). In 1883, Frank reported that after the conidium of Fusicladium sp. and Colletotrichum lindemuthianum germinated on the surface of plants, the tips of the bud tubes expanded to form appressorium (Shang et al., 2021). The production of fungal appressorium directly affects the molecular recognition of pathogenic fungi and host plants, as well as the invasion of pathogenic fungi and the occurrence of host plant defense response (Dar & Mahajan, 2023). The mycelium of many plant parasitic fungi grows on the surface of the host plant, from which collateral branches invade the host cell to absorb nutrients, which is called haustorium. The parasitism of Erysiphales stop and Uredinomycetes sap in plants is characterized by the formation of phagocytes (Polonio et al., 2021a). The outer membrane of the haustorium, known as the haustorium outer membrane, serves as the interface between the haustorium and the host cell plasma membrane (Polonio et al., 2021b). This change is believed to facilitate the passive diffusion of nutrients from the host cell towards the haustorium (Liu et al., 2021).

Referred to the DDRT-PCR technique using long primers and the above two-step PCR amplification as Enhanced Diff-ferential Display (Song et al., 2023; Christensen & Olsen, 2023). The DDRT-PCR results indicated a more pronounced gene expression in infected A. crispa (Thunb.) A.DC. Since Fridovich proposed the theory of biological free radicals, the relationship between the cellular protective enzyme system and plant disease resistance has garnered widespread attention (Hassan, 2022). The zinc finger protein, a superfamily protein with a typical domain, was deemed to be associated with the regulation of plant growth, development, and stress adaptation, and plays a crucial role in regulating plant adaptation to stressful environments. Anion channels have the ability to regulate plant growth, development, and nutrient absorption, making them essential for plants to cope with drought, salt stress, aluminum toxicity, and phosphorus deficiency. The regulatory subunit can influence the activity of the catalytic subunit by binding to an effector and altering its conformation. Carboxylesterase (CXE), a supergene family of polymeric proteins, is primarily engaged in the hydrolysis of restricted substances in plants, such as esters, sulfates, and amides. It plays a major role in plant growth and developmentl. In the study of signal transduction of plant immune response, 26 S proteasome plays a role in the regulation of protein degradation by numerous important transducers (Livneh et al., 2016). The discovery of these differential genes preliminarily elucidated the molecular mechanism of sooty blotch.

Conclusions

This research reports, for the first time, the presence of Annulohypoxylon stygium and D. angeliae as pathogenic fungi affecting A. crispa (Thunb.) A.D. sooty blotch, and the Lonicera fulvotomentosa Hsu et S. C. Cheng and A. crispa (Thunb.) A.DC. had strong antifungal action on two pathogenic fungi of sooty blotch. It can provide a theoretical foundation for the subsequent green prevention and control technology of A. crispa (Thunb.) A.DC. sooty blotch, and also offer a guarantee and theoretical support for the sustainable and healthy development of the A. crispa (Thunb.) A.DC. planting industry.

Supplemental Information

Supplemental Information 1 The result of RNA extraction.

Supplemental Information 2 Polyacrylamide gel electrophoresis.

Supplemental Information 3 DNA recovery from different bands by agarose gel electrophoresis.

Supplemental Information 4 Single bands.

Supplemental Information 5 Disease morphology of the two pathogenic fungi in different periods.

The colony status of healthy leaf of Ardisia crispa (Thunb.) A.DC. was infected with pathogenic fungi for 3 days, 8 days and 12 days.

Supplemental Information 6 Supplemental Tables.

Additional Information and Declarations

Competing Interests

The authors declare that they have no competing interests.

Author Contributions

Demei Yang performed the experiments, prepared figures and/or tables, authored or reviewed drafts of the article, contributed reagents, and approved the final draft.

Jiangli Luo performed the experiments, authored or reviewed drafts of the article, and approved the final draft.

Ying Zhou conceived and designed the experiments, authored or reviewed drafts of the article, and approved the final draft.

Sixuan Zhou analyzed the data, authored or reviewed drafts of the article, and approved the final draft.

Xiongwei Liu analyzed the data, prepared figures and/or tables, authored or reviewed drafts of the article, and approved the final draft.

Chang Liu conceived and designed the experiments, prepared figures and/or tables, authored or reviewed drafts of the article, contributed materials, analysis tools, and approved the final draft.

Data Availability

The following information was supplied regarding data availability:

The raw data are available in the Supplemental Files.

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
