# Peer review of "Identification and biological characterization of pathogen causing sooty blotch of Ardisia crispa (Thunb.) A.DC"

_PeerJ, doi:10.7717/peerj.19130_

## Round 0.1 · original submission · Major Revisions

Dear authors,
Please revise according to the reviewer's comments.

·

Basic reporting

1. Language and Clarity:
o The manuscript is written in professional English, but several instances require grammar and phrasing improvements for clarity (e.g., lines 16-18, 67-70, 79-81). Suggest the authors seek language editing to enhance comprehension.
o Example: "The study elucidates molecular mechanisms of biological action from the perspective of genes" (line 81) could be rephrased for clarity.

2. Introduction and Context:
o The introduction provides relevant background and establishes the importance of the study. However, the authors could elaborate further on the knowledge gap and why this specific fungal characterization is novel.
o Suggest expanding the discussion on the global relevance of A. crispa as a medicinal plant.

3. Figures and Tables:
o Figures are generally clear and well-labeled. However, the resolution of Figures 1 and 2 could be improved for better visualization of microscopic observations.
o Tables (e.g., Table 1-3) provide useful data but require captions clearly stating their purpose and relevance to the text.
o For example, suggestion caption for Table 1 > Primer sequences used for 3' end anchoring in DDRT-PCR for analysing gene expression differences between healthy and diseased Ardisia crispa leaves.

4. Raw Data Availability:
o Raw data is provided, but a detailed explanation of how it can be accessed or reused is missing (please ignore this if not relevant).

Experimental design

1. Scope and Originality:
o The study falls within the journal's scope, addressing the novel identification of two fungal pathogens affecting A. crispa.
o The research question is well-defined and meaningful, providing new insights into fungal pathogens in medicinal plants.

2. Methodology:
o The methodologies are detailed, but some steps could be clarified for reproducibility:
 Line 150: Expand on the criteria used to distinguish fungal strains morphologically and genetically.
 Include more specifics about the DDRT-PCR primer selection process.

3. Field Study Permits:
o Ensure that the authors provide explicit documentation of field study permits, as required.

Validity of the findings

1. Data Presentation:
o Data is statistically sound and presented logically. However, a clearer distinction between the results for Annulohypoxylon stygium and Diaporthe angelicae would enhance understanding.
o Provide more robust statistical analysis, particularly for antifungal efficacy comparisons.

2. Conclusions:
o The conclusions are generally supported by the results but should better highlight the implications of the findings for fungal management in medicinal plants.

Additional comments

I think:

1. The study is valuable, particularly for understanding fungal pathogens of medicinal plants.
2. Highlighting broader applications of this research for sustainable agriculture and biocontrol strategies could enhance the manuscript's impact.
3. Suggest adding a separate section or paragraph in the discussion on limitations and potential follow-up studies.

Reviewer 2 ·

Basic reporting

I believe the literature references provided are sufficient in the background/context section. However, in the discussion, particularly in lines 328-329, I suggest using more recent references, ideally from the year 2000 onward. Additionally, the last part of the discussion lacks references to support the arguments presented between lines 360 and 402. On the other hand, I believe that Figure 5 is not relevant to the paper, as it does not show either the microscopic or macroscopic characteristics of the pathogenic fungi causing sooty blotch. I also consider it important to include the quantitative results of the inhibitory effects of the different extracts presented in Tables S7 and S8 in an acceptable format.
Finally, I believe that part of the results is not fully aligned with one of the main statements. The authors themselves mention in the discussion that sooty blotch does not invade the plant's inner tissues and does not directly destroy the plant. Instead, it indirectly affects the plant's physiological functions by reducing the effective photosynthetic area of the leaves. However, in the results, specifically in Figure 1B, the authors indicate that a significant number of pathogenic fungi were observed within the triangular area, and some areas also showed disrupted cell membranes. This raises the question: are the pathogenic fungi identified truly responsible for the development of sooty blotch?"

Experimental design

I believe that some of the methods described lack sufficient detail to allow replication. Specifically, in the section on the biological control of pathogenic fungi, the authors mention using eight types of Chinese medicine extracted through reflux heating. However, they do not specify the solvent used for the extraction

Validity of the findings

No comment.

Additional comments

In the attached file, I marked some notes with different colors: Yellow indicates that it is necessary to separate words, such as the value from the unit (e.g., '50 mL' instead of '50mL'). Green indicates that the word should be written in italics, for example, the scientific names of organisms. Blue highlights orthographic mistakes that need correction, such as writing 'pH' instead of 'PH.' Additionally, units, particularly volume units like 'L,' should be standardized.

Annotated reviews are not available for download in order to protect the identity of reviewers who chose to remain anonymous.

---

## Round 0.2 · Minor Revisions

Please note comments in the annotated manuscript

·

Basic reporting

no comment

Experimental design

no comment

Validity of the findings

no comment

---

## Round 0.3 · Major Revisions

Dear authors,
Before making a decision on the acceptance of this paper, I would like to invite you to improve the statistical component of the study. In methodology, please state which post-hoc test you used to differentiate levels of the factors in your ANOVA. Also, please state how did you check the prerequisites of the ANOVA. In results, particularly in Tables 1 and 2, again state the post-hoc test used, and clarify if the means are presented with a SD or if these are CI. Also, add notes to inform the reader about the superscripts.

---

## Round 0.4 · Major Revisions

Dear authors,

Your answers to my questions show a lack of expertise in statistics. I urge you to seek statistical advice from an expert and reformulate your answers. Please pay close attention to all the issues raised. This paper can not be accepted if the statistical component is not reported according to the standards set in instructions for authors.

---

## Round 0.5 · accepted · Accept

The statistical issues raised were now properly addressed. I am, therefore, happy to recommend the publication of this paper.